# The Upper Bound Theorem in Forging Processes: Model of Triangular Rigid Zones on Parts with Horizontal Symmetry †

Francisco Martín * , María Jesús Martín and María José Cano

Civil, Material and Manufacturing Engineering Department, EII, University of Málaga, 29071 Málaga, Spain; mjmartin@uma.es (M.J.M.); mjcano@uma.es (M.J.C.)
* Correspondence: fdmartin@uma.es; Tel.: +34-951-952-309
† This paper is the extended version of the conference paper published in the 8th Manufacturing Engineering Society International Conference, MESIC 2019, Madrid, Spain, 19–21 June 2019.

**Abstract:** This paper presents the analytical method capacity of the upper bound theorem, under modular approach, to extend its application possibilities. Traditionally, this method has been applied in forging processes, considering plane strain condition and parts with double symmetry configuration. However, in this study, the double symmetry is eliminated by means of a fluency plane whose position comes from the center of mass calculated. The study of the load required to ensure the plastic deformation will be focus on the profile of the part, independently on both sides of the fluence plane, modifying the number and the shape of the modules that form the two halves in which the part is defined. This way, it is possible to calculate the necessary load to cause the plastic deformation, whatever its geometric profile.

**Keywords:** upper bound; plane strain; forging; triangular rigid zones





## 1. Introduction

The high complexity of the theory of plasticity has conditioned its analytical development. One of the major limitations to determine mathematical relation that allows us to know the initial conditions of the plastic deformation of a material comes from the irreversible and nonlinear character of this type of deformation. With this objective, several families of methods have been developed in the study on metal alloys of conformation processes by plastic deformation (PCPD) [1,2], and, especially, in forging processes, applying both numerical and analytical methods.

Analytical methods increase the ability of the engineer to evaluate and predict the influence of certain variables on important aspects of the process, for example, the necessary energy to achieve it. The first methods among that proposed are based on simple theoretical fundaments, in which only the geometrical aspects of the part are considered, as well as the distribution of the tension in the plastically deformed area. Such methods are the homogeneous deformation method and the local stress analysis methods. Both of them have the advantage of their relative simplicity of application in comparison with other methods used in the PCPD analysis. Unlike the two previous methods, the sliding lines field method (SLD) [3,4] presents a methodological alternative whose complexity in its application is directly related to the accuracy level required to solve the problem, since it is based on the definition of a yield field which is itself difficult to define.

The complexity of the mathematical development of the equations restricts the study to processes in plane strain conditions, as well as to reduced complexity geometries. However, the bound analysis methods and, more specifically, the upper bound theorem [5,6], are supposed to be a solid alternative to the application of the SLD method. This method can be considered a particular case of the SLD but is easier to implement, and provides quite acceptable solutions, nevertheless. At present, the high computational capacity of computers has led to a high development of numerical methods, among which is the finite

elements method, displacing these analytical methods. However, the conditions of simplification that the bound analysis methods offer place them in a situation of effectiveness similar to the numerical ones mentioned before.

In the present article, the upper bound method is approached through a triangular rigid zones model [7]. The exact solutions for plastic deformation problems are difficult to obtain. According to the limit theorems, an approximation to them is to define the solution for the necessary energy of deformation between the lower and upper limits. The power must break the resistance of the material to the deformation, as well as the resistance to displacement, this latter due to the friction between the material and the tool. The real load will be delimited between these upper and lower limits, although the first one will be more interesting, since it ensures that the deformation can be carried out by the calculated load. This method has important advantages for the determination of particular solutions.

The upper bound criterion applies the principle of maximum work but, from the point of view of deformation, that is, an element is deformed in such a way as to offer the minimum resistance. When deducting a stress system from a hypothetical deformation that is in accordance with the kinematic conditions, the load value obtained will be greater or equal to the one that actually operates. When establishing the appropriate deformation, it will be necessary to define a kinematically admissible velocity field, independently of the tensional conditions, which is usually represented thorough its hodograph. The mathematical formulation of the upper bound theorem (UBT) presented by Prager and Hodge [8] indicates that, taking into account the discontinuity surfaces between the different rigid blocks considered, the actual kinematically admissible velocity field among the possible velocity fields is that one that minimizes the following expression:

$$\int_{S_v} T_i \, v_i dS_v \ \leq \ \int_{S_D} k \, [v^*] \, dS_D^* + \int_{S_F} Ti \, v_i^* \, dS_F \tag{1}$$

where $T_i$: external surface stresses on the workpiece to form; $v_i$: actual velocity field; $S_v$: surfaces where the external forces are applied; $k$: shear yield stress; $v^*$: velocity discontinuities; $S^*_D$: discontinuity surfaces; $v^*_i$: kinematically admissible virtual velocity field; $S_F$: external surfaces exposed to external surface stresses.

For this, it is useful to divide the deformed part into several zones, with a rigid behavior and which are called triangular rigid zones (TRZ) (Kudo [9–13]), in each of which the velocity field and its derivatives must be continuous. The application of this method is done by straight lines, considering that only along them there are velocity discontinuities (Rubio [14–16], Qin [17], and Yang [18]). The rest of the points that make up each block move at the same speed and with the same direction.

In a particularly unique way, and with reduced technological changes, the forging processes are fully adapted to the application of this method. Modifications are aimed at achieving a flow behavior of material very similar to that of flat deformation, which is achieved by designing pieces of straight generatrix in the plane perpendicular to the one under study.

These TRZ allow the incorporation of different variables present in the plastic forming of metal alloys. In this way, and given that the forge constitutes a nonstationary process, it is possible to determine different natures of the friction (Tresca or Coulomb [19–21]) acting on the different flat surfaces of contact between part and tool, and, even, assigning different values to each surface. Other parameters, such as the temperature of the process, also have the possibility of being analyzed.

One of the main limitations in the classic application of the UBT by means of TRZ comes from the imposition of the double symmetry, although different approaches have been proposed to relative nonsymmetrical parts [22–24] that prevents us from considering a large number of the geometric configurations present in the industry. In the present work, one of these symmetries has been eliminated, thus significantly increasing the range of application of the method.

Consequently, compared with the two options (complex and numerical analytical methods with high computational cost), the analytical method based on the UBT is presented as an interesting alternative to calculate the minimum load to reach the permanent deformation in forging processes, considering a plane strain state. This method offers a reasonable rough value of the necessary load (in terms of pressure) to guarantee the plastic deformation of the part. The application of the UBT, using the model of triangular rigid zones, gives solutions with an extremely low computational cost, since it is possible to develop it immediately by using a simple spreadsheet.

## 2. Methodology

The limit analysis (lower and upper) provides analytical solutions that delimit the range where the solution must exist. The UBT is one analytical method that can provide quite accurate results when it is properly applied. Among the two approaches of limit analysis, the UBT is chosen because it is not so difficult to apply and also because of its theoretical fundamentals, where the deformation is guaranteed for the obtained solution.

The UBT not only provides the minimum value of load that assesses the desired deformation but also allows to define the involved parameters so optimal process conditions can be determined.

In the traditional application of the method, the profile of the part was configured by means of a series of triangular rigid zones which increased in number and shape with the result of equations more and more complex. The geometrical parameters of each TRZ configuration and their hodograph are represented in Figures 1 and 2 for an odd and an even number of TRZ to study the influence of the TRZ number in the $P/2k$ relation, where $p$ is the mean forging pressure and $k$ is the shear yield stress of this material.

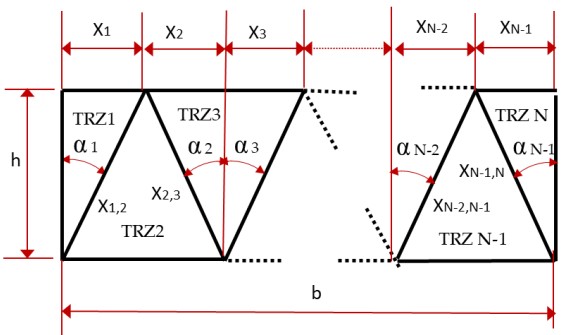 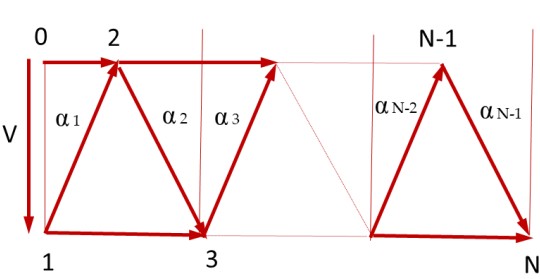

**Figure 1.** Distribution and hodograph with odd number of triangular rigid zones (TRZ).

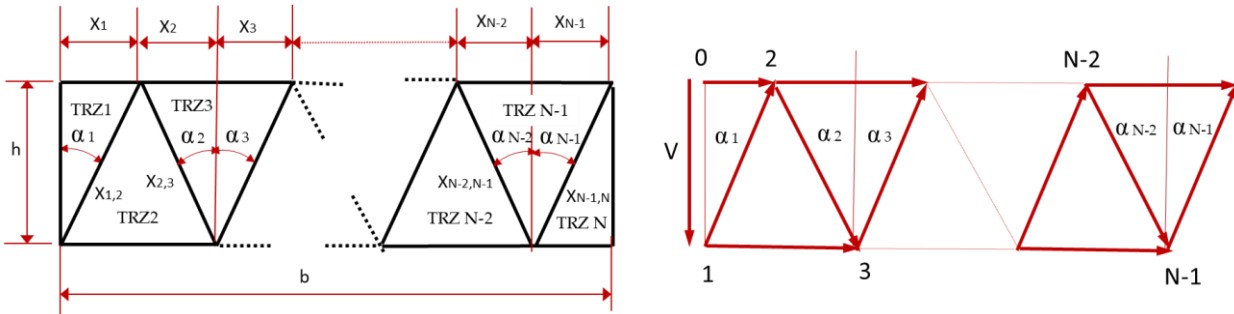

**Figure 2.** Distribution and hodograph with even number of TRZ.

The application of the UBT in each of these four cases should obtain optimized values of $x_i$ parameters in different boundary conditions.

The consideration of a semi-adherence in die–workpiece contact has been solved including an *m* friction shear factor in affected terms of Equation (16).

The *p*/2*k* relation is evaluated according to shape factor over a variable number of TRZ, obtaining a generalization for *n* TRZ expression. Equations (2) and (3) show the deformation power expressions for an odd number (Equation (2)) and an even number (Equation (3)) of TRZ:

$$\frac{dW}{dt} = \dot{W} \leq kw \left[ \sum_{i=1}^{N-1} v_{i,i+1} x_{i,i+1} + m \sum_{i=1}^{(N-1)/2} v_{1,2i+1}(x_{2i} + x_{2i+1}) \right]; \ x_N = 0 \quad (2)$$

$$\frac{dW}{dt} = \dot{W} \leq kw \left[ \sum_{i=1}^{N-1} v_{i,i+1} x_{i,i+1} + m \sum_{i=1}^{N/2-1} v_{1,2i+1}(x_{2i} + x_{2i+1}) \right] \quad (3)$$

Applying the trigonometric ratios for the positions and velocities of each zone is possible to simplify these equations, obtaining Equation (4) for an odd number of TRZ

$$\frac{dW}{dt} = \dot{W} \leq \frac{kwv}{h} \left[ h^2(N-1) + \sum_{i=1}^{N-1} x_i^2 + m \sum_{i=1}^{(N-1)/2} \left( \sum_{j=1}^{2i} x_j \right)(x_{2i} + x_{2i+1}) \right]; \ x_N = 0 \quad (4)$$

and Equation (5) for an even number of TRZ.

$$\frac{dW}{dt} = \dot{W} \leq \frac{kwv}{h} \left[ h^2(N-1) + \sum_{i=1}^{N-1} x_i^2 + m \sum_{i=1}^{(N-2)/2} \left( \sum_{j=1}^{2i} x_j \right)(x_{2i} + x_{2i+1}) \right] \quad (5)$$

As shown, this approach only considered plane-parallel surfaces, increasing the terms of the equation according to the number of TRZ defined.

Following previous works developed by the authors of this paper, this study establishes a new approach in which the basic equation develops from the geometrical configuration of a generic module of three TRZ [25–27]. This module can connect to subsequent ones through the yield velocities of the material, being the output velocity from the first module, $V_s$, the input velocity to the next module, $V_e$. The nondimensional relation *P*/2*k* (*p* = pressure on the profile; *k* = shear yield strength of the material) will be always the same, whatever the module.

This new approach, called modular approach by the authors, provides an extension of the upper bound application to geometric configurations in which the tool profile of the flat die combines parallel and inclined flat surfaces. The most important limitation of the UBT by means of TRZ model is the fact that a double symmetry in geometries subjected to a plane strain state is imposed. In this study, one of these boundary conditions is released, allowing to expand the possibilities of application of this method.

The first studies on this topic were developed under plane strain conditions by Kudo [5], creating one field of virtual deformation of the material and comprising rigid blocks, not strictly triangular, but keeping as far as possible the geometric regularity for all of them. This fact restricted the possibility of adapting the rigid block to the whole system that is being deformed. Further studies [9] state different approaches, imposing a variable number of TRZ which depends on the geometric ratio of the width *b* and the height *h* of the quarter of workpiece submitted to analysis; or even choosing different triangular shapes among them, trying to optimize the searched limit.

The advance of this work on the modular approach is based on the horizontal symmetry (to the vertical plane) of the profile. Following this idea, two profiles from pieces with double symmetry are analyzed independently. From these analyses, a quarter of each case will be extracted, both profiles being combined into a new one where the symmetry has been eliminated. In this new profile will be studied the x-coordinate of the center of mass. In this position will be situated the new fluence plane and from it, the material will flow in opposite directions. This way, the modules configuration will be modified with

respect to the previously existing one. Afterward, a new application of the method will be performed, obtaining a resultant modular configuration.

The basic module on which the appropriate combination that represents the initial profile of the section of the part will be established is formed by three TRZ, and is considered under the modular approach (Figure 3a). The basic module will respond in the evolution of the fluence of the material contained in it with the determination of an output velocity of the module from the input velocity and geometric characteristics. This evolution can be observed in its corresponding hodograph (Figure 3b).

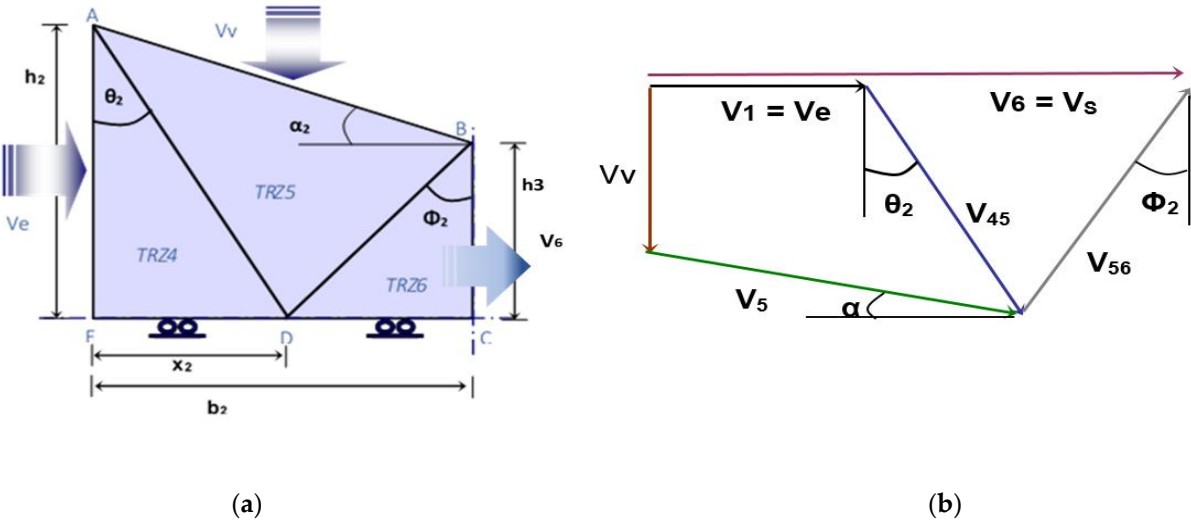

(**a**)                                                                                   (**b**)

**Figure 3.** (**a**) Configuration of the basic module; (**b**) hodograph.

Being a basic module, it must have a general character, so it is necessary that it be defined with all the forces and speeds susceptible to occurrence, that is, input and vertical velocities, and also have the possibility of the previous module, in the case that it exists. This previous module would be fixed on the vertical plane of symmetry (if it is an initial module, the input velocity will be zero and will cancel out everything that is associated with it). This is why the denomination of zones has been used as 4, 5, and 6. (TRZ1, TRZ2, and TRZ3 would correspond to the initial modulus, arranged on the plane that delimits the flow of the material). The position of the lower vertex of the TRZ5 is not fixed in a first analysis and, therefore, the parameter $x_2$ appears.

From the geometric configuration of the basic module:

$$\overline{AD} = \frac{x_2}{sen\theta_2}; V_{45} = \frac{V_1 + V_5 sen\alpha_2}{cos\theta_2}; \overline{DB} = \frac{b_2 - x_2}{sen\varphi_2}; V_{56} = \frac{V_1 + V_5 sen\alpha_2}{cos\varphi_2}; \overline{AB} = \frac{b_2}{cos\alpha_2} b_2; V_5 = \frac{V_1\left(tg\theta_2 + \frac{tg\theta_1 + tg\varphi_1}{1 - tg\alpha_1 tg\theta_1}\right)}{cos\alpha_2 - sen\alpha_2 tg\theta_2} \tag{6}$$

Being the input velocity from a previous module:

$$Ve = V_1 \cdot \left(\frac{tg\theta_1 + tg\varphi_1}{1 - tg\alpha tg\theta_1}\right) \tag{7}$$

and:

$$\frac{V_{56}}{sen\left(\frac{\pi}{2} - \theta_2\right)} = \frac{V_{45}}{sen\left(\frac{\pi}{2} - \varphi_2\right)} = \frac{V_6 - V_1 \cdot \left(\frac{tg\theta_1 + tg\varphi_1}{1 - tg\alpha tg\theta_1}\right)}{sen(\theta_2 + \varphi_2)} \tag{8}$$

The output velocity $V_6$ will be:

$$V_6 = Ve + V_{45}sen\theta_2 + V_{56}sen\varphi_2 = Ve + V_{45}sen\theta_2 + V_{45}\frac{cos\theta_2}{cos\varphi_2}sen\varphi_2 = Ve + V_{45}(sen\theta_2 + cos\theta_2 tg\varphi_2) \tag{9}$$

$$V_6 = V_1\left[\left(\frac{tg\theta_1 + tg\varphi_1}{1 - tg\alpha tg\theta_1}\right) + \left(1 + \frac{\left(tg\theta_2 + \frac{tg\theta_1 + tg\varphi_1}{1 - tg\alpha tg\theta_1}\right)\cdot tg\alpha_2}{1 - tg\alpha_2 tg\theta_2}\right)(tg\theta_2 + tg\varphi_2)\right] \tag{10}$$

where $V_1 = Ve$ is the input velocity to each module.

Applying the UBT will calculate the dimensionless ratio $P/2k$ to obtain the value of the final load.

$$\frac{dW}{dt} = \dot{W} = k \cdot \omega \cdot \left[\overline{AD}\cdot v_{45} + \overline{DB}\cdot v_{56} + m\cdot\overline{AB}\cdot v_5\right] = P\cdot\omega\cdot b_2\cdot V_1 \tag{11}$$

and considering that:

$$tg\theta_2 = \frac{x_2}{h_2}; tg\theta_1 = \frac{x_1}{h_1}; tg\varphi_1 = \frac{b_1 - x_1}{h_2}; sen\theta_2 = \frac{x_2}{\sqrt{x_2^2 + h_2^2}}; cos\theta_2$$
$$= \frac{h_2}{\sqrt{x_2^2 + h_2^2}}; sen\varphi_2 = \frac{b_2 - x_2}{\sqrt{h_3^2 + (b_2 - x_2)^2}}; cos\varphi_2$$
$$= \frac{h_3}{\sqrt{h_3^2 + (b_2 - x_2)^2}}; \tag{12}$$

we obtain:

$$\dot{W} = k\cdot\omega\cdot\left[\frac{x_2}{sen\theta_2}\frac{V_1 + V_5 sen\alpha}{cos\theta_2} + \frac{(b_2 - x_2)}{sen\varphi_2}\frac{V_1 + V_5 sen\alpha_2}{cos\varphi_2} + m\cdot\frac{b_2}{cos\alpha_2}\cdot\frac{V_1\cdot\left(tg\theta_2 + \frac{tg\theta_1 + tg\varphi_1}{1 - tg\alpha tg\theta_1}\right)}{cos\alpha_2 - sen\alpha_2 tg\theta_2}\right] \tag{13}$$

The modular approach has been implemented with the possibility of incorporating friction both by adhesion (Tresca, *m*) (Equation (14)) or sliding (Coulomb, *μ*) (Equation (15)) [16], responding in each case to a different expression.

$$\frac{P}{2k} = \frac{1}{2b_2}\cdot\left[\left(\frac{x_2^2 + h_2^2}{h_2} + \frac{(b_2 - x_2)^2 + h_3^2}{h_3}\right)\left(1 + \frac{\left(tg\theta_2 + \left(\frac{tg\theta_1 + tg\varphi_1}{1 - tg\alpha tg\theta_1}\right)\right)}{cos\alpha_2 - sen\alpha_2 tg\theta_2}sen\alpha_2 + \frac{m\cdot b_2}{cos\alpha_2}\right)\cdot\frac{\left(tg\theta_2 + \left(\frac{tg\theta_1 + tg\varphi_1}{1 - tg\alpha tg\theta_1}\right)\right)}{cos\alpha_2 - sen\alpha_2 tg\theta_2}\right] \tag{14}$$

$$\frac{P}{2k} = \frac{\left(\frac{x_2^2 + h_2^2}{h_2} + \frac{(b_2 - x_2)^2 + h_3^2}{h_3}\right)\cdot\left(1 + \frac{\left(tg\theta_2 + \frac{tg\theta_1 + tg\varphi_1}{1 - tg\alpha tg\theta_1}\right)\cdot sen\alpha_2}{cos\alpha_2 - sen\alpha_2 tg\theta_2}\right)}{2b_2\cdot\left(1 - \frac{\mu}{cos\alpha_2}\frac{\left(tg\theta_2 + \frac{tg\theta_1 + tg\varphi_1}{1 - tg\alpha tg\theta_1}\right)}{cos\alpha_2 - sen\alpha_2 tg\theta_2}\right)} \tag{15}$$

The combined geometry is analyzed in a modular way, stating that the upper bound of the power can be obtained as a result of the combination of modules with different geometry; according to the law of continuity, the exit velocity of the material that comes from the first module has to be equal to the entrance velocity of the material in the second module (Figure 4).

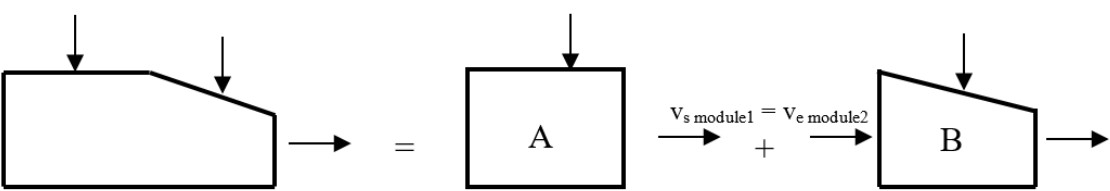

**Figure 4.** Module velocities.

As stated above, the connection between modules is defined by the exit velocity from one module that is the same than the entrance velocity to the next one, denoted by $V_e$, and whose value can be determined by Equation (16).

$$V_6 = V_1 \left[ \left( \frac{tg\theta_1 + tg\varphi_1}{1 - tg\alpha tg\theta_1} \right) + \left( 1 + \frac{\left( tg\theta_2 + \frac{tg\theta_1 + tg\varphi_1}{1 - tg\alpha tg\theta_1} \right) \cdot tg\alpha_2}{1 - tg\alpha_2 tg\theta_2} \right) (tg\theta_2 + tg\varphi_2) \right] \quad (16)$$

The calculation of the ratio $P/2k$ combining the influence of both modules ($P_{A+B}/2k$) is determined by Equation (17), where some weight factors are included, considering the geometry of each module by means of the width.

$$\frac{P_{A+B}}{2k} = \frac{\frac{P_A}{2k} \cdot b_1 + \frac{P_B}{2k} \cdot b_2}{(b_1 + b_2)} \quad (17)$$

## 3. Results

We initially consider two parts of profiles shown in Figure 5a,b.

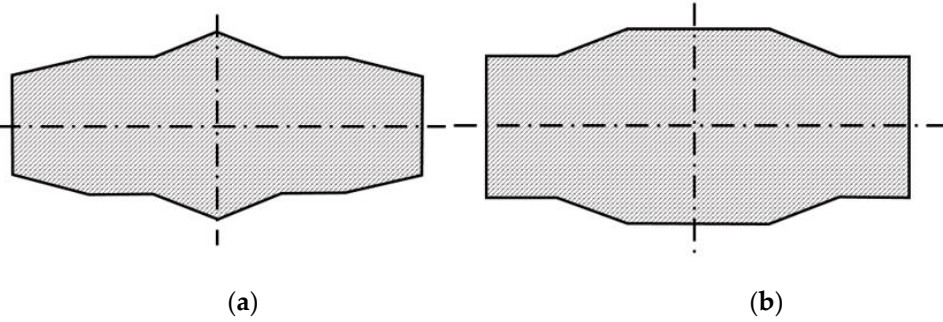

(**a**)　　　　　　　　　　　　　　　　　　(**b**)

**Figure 5.** (**a**) Pieza 1; (**b**) Pieza 2.

From these parts will be adjusted the TRZ configurations and will be studied, from the planes of symmetry, a quarter of each profile, considering the situation of a piece with symmetric configuration (Figures 6 and 7).

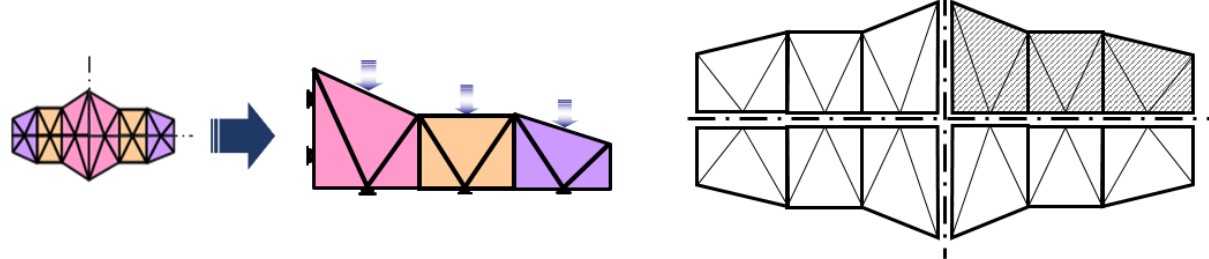

**Figure 6.** 3 TRZ part configuration 1.

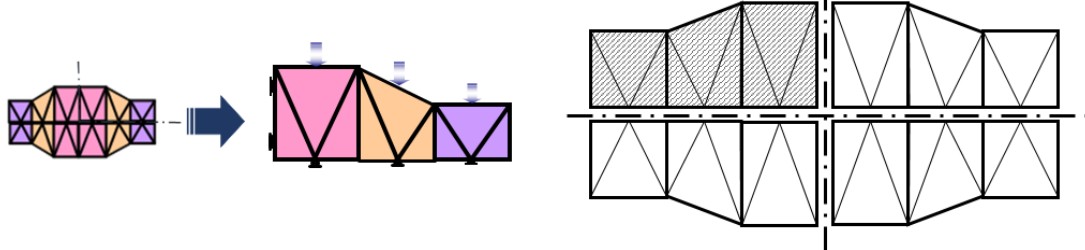

**Figure 7.** 3 TRZ part configuration 2.

Each one of the three TRZ of the selected modules in configuration 1 has the width of 4, 4, and 3, respectively (counted from left to right from the vertical plane of symmetry), with an initial module height of 6. For the piece with configuration 2, the TRZ are 4, 3, and 5 in width, respectively, keeping the same height for the initial TRZ.

Once established the configurations of the two quarters of part selected, both configurations are combined as an only part, generating the profile shown in the next figure (Figure 8).

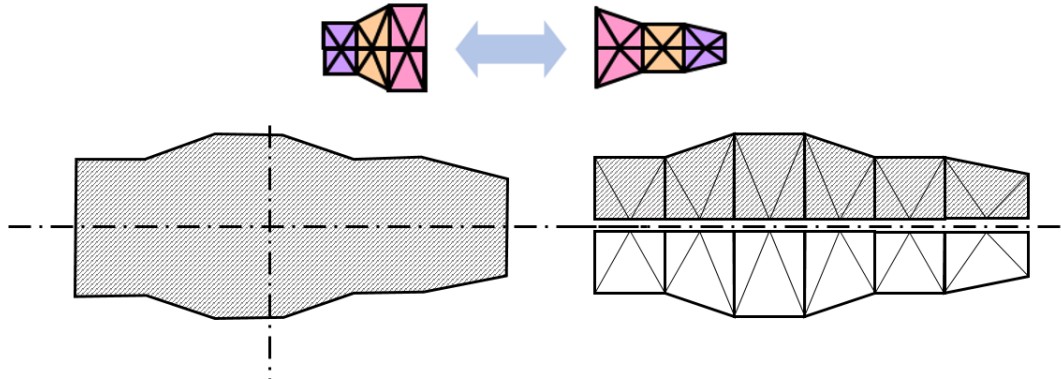

**Figure 8.** New configuration without horizontal symmetry (ws).

As shown, the yield plane of the material derived from the forging process is coincident with the symmetry plane in the initial configurations. In this new configuration will be necessary to calculate the position of the vertical plane from which the material flows in opposite directions. This position will be calculated determining the position of the center of mass of the entire profile on the $X$-axis, taking the point $A$ as origin (Figure 9). In the current configuration, the $X$ dimension is 11.222.

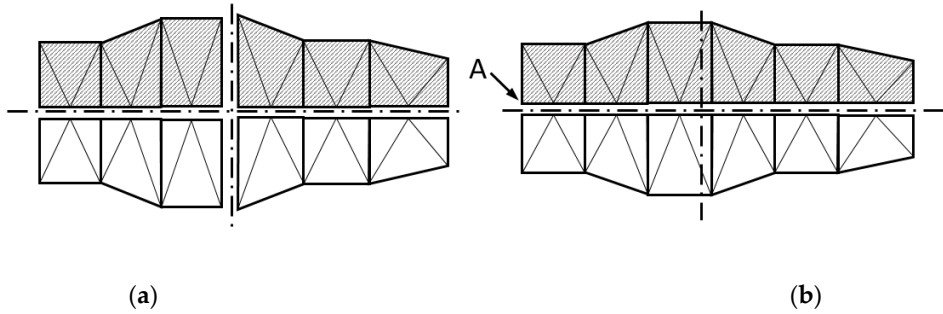

(**a**)  (**b**)

**Figure 9.** Symmetry planes: (**a**) initial configuration; (**b**) new configuration.

After establishing the new symmetry plane, this one intersects one of the modules, so the new resultant configuration has a new additional module of 0.778 in width on the right side of the profile (Figure 10). Now, two new geometric configurations will be generated, situated on the left (configuration 1ws) and on the right (configuration 2ws) with respect to the new yield plane.

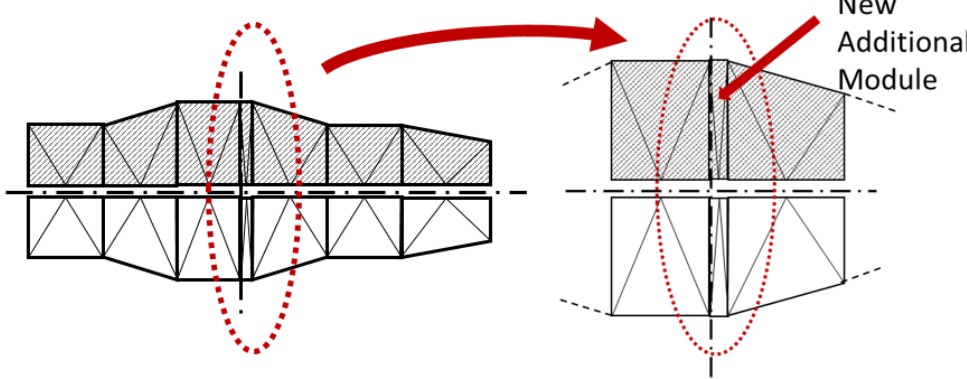

**Figure 10.** New additional module: new configurations 1ws and 2ws.

Once the forging process begins, the height of the section decreases, resulting in the deformation of the part. The *P/2k* ratio is calculated on the different stages of this evolution, from the initial geometry to a high degree of deformation in the two configurations (configuration 1: Figure 11a and Table 1; configuration 2: Figure 11b and Table 2).

**Table 1.** *P/2k* modules values and total values: configuration 1.

| h1 | Module 1 | Module 2 | Module 3 | Total |
|----|----------|----------|----------|-------|
| 6 | 1.683 | 2.359 | 1.432 | 2.166 |
| 5.8 | 1.640 | 2.300 | 1.403 | 2.124 |
| 5.6 | 1.596 | 2.241 | 1.375 | 2.082 |
| 5.4 | 1.554 | 2.183 | 1.348 | 2.042 |
| 5.2 | 1.511 | 2.126 | 1.322 | 2.001 |
| 5 | 1.470 | 2.070 | 1.298 | 1.962 |
| 4.8 | 1.429 | 2.014 | 1.275 | 1.923 |
| 4.6 | 1.390 | 1.960 | 1.253 | 1.855 |
| 4.4 | 1.350 | 1.907 | 1.233 | 1.849 |
| 4.2 | 1.312 | 1.855 | 1.216 | 1.814 |
| 4 | 1.275 | 1.806 | 1.201 | 1.781 |

**Table 2.** *P/2k* modules values and total values: configuration 2.

| h1 | Module 1 | Module 2 | Module 3 | Total |
|----|----------|----------|----------|-------|
| 6 | 1.699 | 1.611 | 2.388 | 2.346 |
| 5.8 | 1.657 | 1.570 | 2.333 | 2.300 |
| 5.6 | 1.615 | 1.530 | 2.278 | 2.254 |
| 5.4 | 1.573 | 1.490 | 2.225 | 2.208 |
| 5.2 | 1.533 | 1.451 | 2.173 | 2.163 |
| 5 | 1.493 | 1.414 | 2.123 | 2.119 |
| 4.8 | 1.455 | 1.377 | 2.074 | 2.076 |
| 4.6 | 1.417 | 1.342 | 2.027 | 2.034 |
| 4.4 | 1.381 | 1.308 | 1.983 | 1.993 |
| 4.2 | 1.346 | 1.276 | 1.942 | 1.955 |
| 4 | 1.312 | 1.246 | 1.904 | 1.919 |

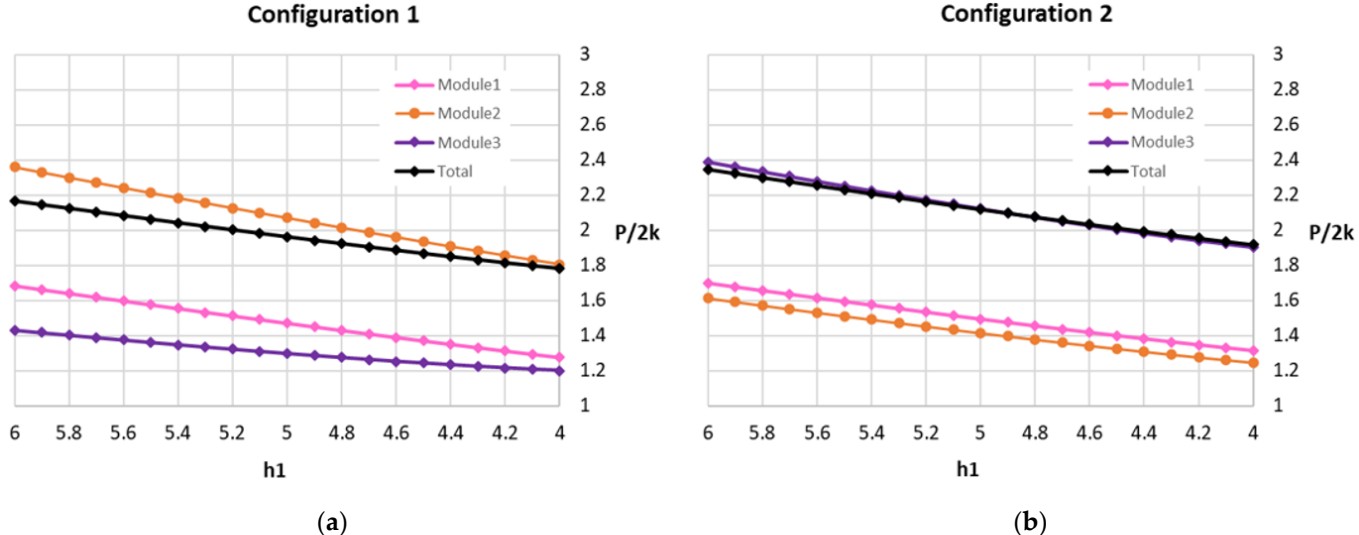

**Figure 11.** (**a**) *P/2k* evolution: configuration 1; (**b**) *P/2k* evolution: configuration 2.

The new configurations 1ws and 2ws are shown (configuration 1ws: Figure 12a and Table 3; configuration 2ws: Figure 12b and Table 4).

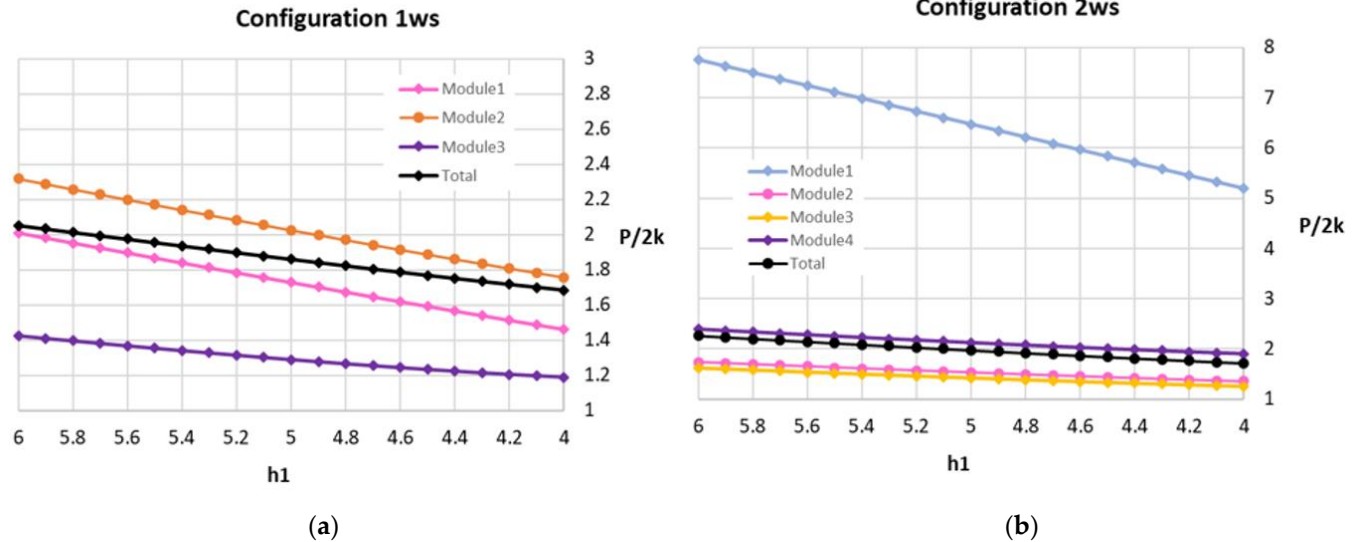

**Figure 12.** (**a**) *P/2k* evolution: configuration 1ws; (**b**) *P/2k* evolution: configuration 2ws.

**Table 3.** *P/2k* modules values and total values: configuration 1ws.

| h1 | Module 1 | Module 2 | Module 3 | Total |
|----|----------|----------|----------|-------|
| 6 | 2.010 | 2.318 | 1.425 | 2.053 |
| 5.8 | 1.953 | 2.258 | 1.396 | 2.014 |
| 5.6 | 1.896 | 2.199 | 1.368 | 1.975 |
| 5.4 | 1.840 | 2.141 | 1.340 | 1.936 |
| 5.2 | 1.784 | 2.083 | 1.314 | 1.898 |
| 5 | 1.730 | 2.026 | 1.289 | 1.860 |
| 4.8 | 1.674 | 1.970 | 1.266 | 1.823 |
| 4.6 | 1.620 | 1.915 | 1.244 | 1.786 |
| 4.4 | 1.567 | 1.861 | 1.224 | 1.751 |
| 4.2 | 1.514 | 1.809 | 1.206 | 1.717 |
| 4 | 1.463 | 1.758 | 1.190 | 1.684 |

**Table 4.** *P/2k* modules values and total values: configuration 2ws.

| h1 | Module 1 | Module 2 | Module 3 | Module 4 | Total |
|----|----------|----------|----------|----------|-------|
| 6 | 7.748 | 1.734 | 1.618 | 2.388 | 2.259 |
| 5.8 | 7.492 | 1.692 | 1.578 | 2.333 | 2.199 |
| 5.6 | 7.236 | 1.650 | 1.537 | 2.278 | 2.141 |
| 5.4 | 6.980 | 1.609 | 1.498 | 2.225 | 2.083 |
| 5.2 | 6.725 | 1.570 | 1.460 | 2.173 | 2.027 |
| 5 | 6.470 | 1.531 | 1.423 | 2.123 | 1.971 |
| 4.8 | 6.214 | 1.493 | 1.386 | 2.074 | 1.917 |
| 4.6 | 5.959 | 1.456 | 1.351 | 2.027 | 1.863 |
| 4.4 | 5.704 | 1.420 | 1.318 | 1.983 | 1.812 |
| 4.2 | 5.450 | 1.386 | 1.287 | 1.942 | 1.762 |
| 4 | 5.915 | 1.355 | 1.257 | 1.904 | 1.715 |

It is interesting to note the new extra module in the configuration 2ws, which comes from the modification of the new yield plane position. Based on the results obtained, it can be observed the influence of the variation of the width of the additional extra module added in configuration 2ws. This module has a very small width, so for module 1, in this configuration, 2ws is highly distorted and could affect the result of the total *P/2k* ratio, increasing it significantly. However, this influence is slightly compensated by the reduced value of the area of the aforementioned module and, therefore, its influence is reduced (Figure 12b, Table 4).

It can be seen how the value of the *P/2k* ratio decreases in the new configurations with the appearance of the new flow plane and the extra module. For example, for a reduction in height up to the value of 6, the values of these relationships in the initial configuration 1 and 2 are 2.166 and 2.346, respectively (Tables 1 and 2), while, with the new redistribution and appearance of a fourth module, in configuration 2ws the resulting values are 2.053 and 2.259 (Tables 3 and 4), with reductions of 0.113 and 0.087 between the initial and final values of our analysis.

Analyzing jointly the entire profiles generated by the configurations 1 and 2, and those obtained with the new combination, 1ws and 2ws, demonstrates how the values resulting from Equation (6) show an upper limit (calculated with the relation *P/2k*) lower when the new yield plane is considered and, therefore, it provides a more accurate adjustment of the upper bound value necessary to make the material flows (Tables 5 and 6, Figure 13a,b).

**Table 5.** *P/2k* values: total configuration 1, configuration 2, and complete profile.

| h1 | Total Conf 1 | Total Conf 2 | Total Profile |
|----|--------------|--------------|---------------|
| 6 | 2.166 | 2.346 | 2.252 |
| 5.8 | 2.124 | 2.300 | 2.208 |
| 5.6 | 2.082 | 2.254 | 2.164 |
| 5.4 | 2.042 | 2.208 | 2.121 |
| 5.2 | 2.001 | 2.163 | 2.079 |
| 5 | 1.962 | 2.120 | 2.037 |
| 4.8 | 1.923 | 2.076 | 1.996 |
| 4.6 | 1.885 | 2.034 | 1.956 |
| 4.4 | 1.850 | 1.993 | 1.918 |
| 4.2 | 1.814 | 1.955 | 1.881 |
| 4 | 1.781 | 1.919 | 1.847 |

**Table 6.** *P/2k* values: total configuration 1ws, configuration 2ws, and complete profile.

| h1 | Total Conf 1ws | Total Conf 2ws | Total Profile ws |
|:--:|:--:|:--:|:--:|
| 6 | 2.053 | 2.259 | 2.158 |
| 5.8 | 2.014 | 2.199 | 2.109 |
| 5.6 | 1.975 | 2.141 | 2.060 |
| 5.4 | 1.936 | 2.083 | 2.012 |
| 5.2 | 1.898 | 2.027 | 1.964 |
| 5 | 1.860 | 1.971 | 1.917 |
| 4.8 | 1.823 | 1.916 | 1.871 |
| 4.6 | 1.786 | 1.863 | 1.826 |
| 4.4 | 1.751 | 1.812 | 1.782 |
| 4.2 | 1.717 | 1.762 | 1.741 |
| 4 | 1.684 | 1.715 | 1.700 |

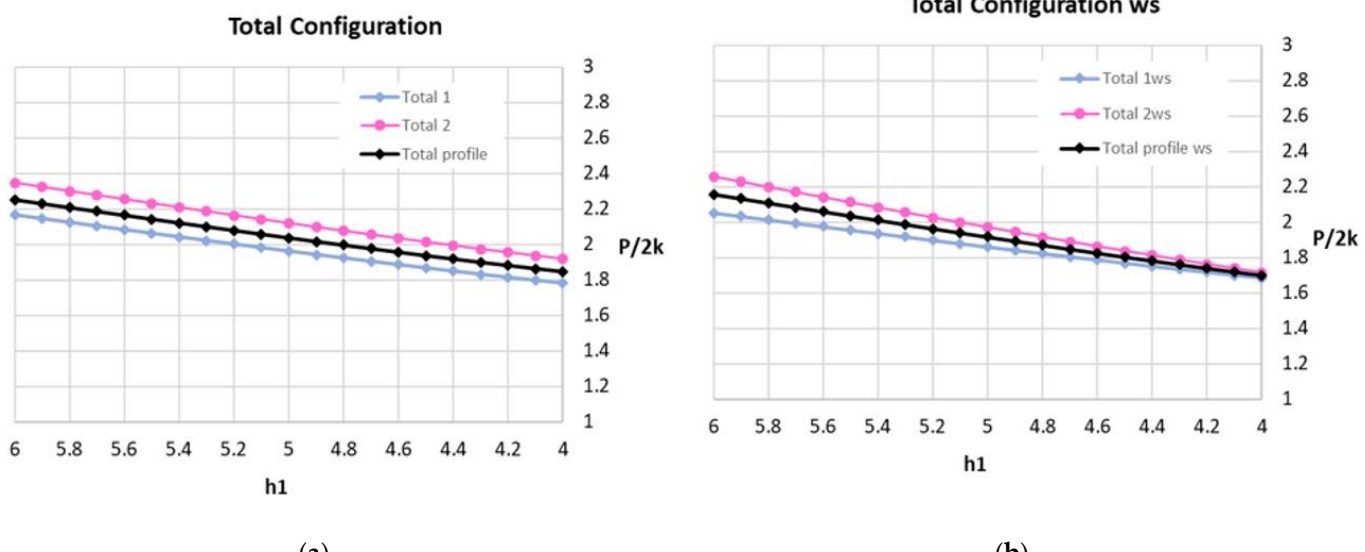

(**a**)                                               (**b**)

**Figure 13.** (**a**) *P/2k* evolution: configuration 1ws; (**b**) *P/2k* evolution: configuration 2ws.

### 4. Conclusions

In addition to the advantages presented by the upper bound theorem by means of triangle rigid zones (TRZ) under modular approach, such as easy implementation, low computational cost, or quick calculation response, this work shows an extension of its possibilities of application. The initial analysis was limited to parts produced by forging with dies of plane-parallel plates. The modular approach allowed, in a simple way, to be used in parts with symmetrical profiles. Finally, this paper extends its application to nonsymmetrical profiles.

Considering a nonsymmetrical profile presents the difficulty of defining the vertical plane from which the material flows in opposite directions, which does not happen in symmetrical parts. In this case, the plane of symmetry is the position from which the yield occurs. Determining the new plane of symmetry requires a redistribution and a reconfiguration of modules to be adapted to the new profile. This ingenious application allows to generate the yield plane and, according to the results obtained, a lower value of the maximum limit calculated, adjusting the final result to the real value of the load required to reach the plastic deformation of the material.

Another conclusion drawn from this study is the fact that the use of very distorted modules (very high relation between height and width) does not mean a significant negative influence, because its low module width provides a very reduced weight in the overall calculation of the nondimensional relation *P/2k*.

The possibilities of application through this triangular rigid zones (TRZ) model remain intact, both in parameters contemplated here, reflected in the theoretical development (shear and Coulomb friction), and in others not presented in this work (temperature or strain hardening), that provide solutions through the application of an extremely low computational cost method of immediate response, strongly competitive with numerical methods.

**Author Contributions:** Conceptualization and investigation: M.J.M., F.M., and M.J.C.; methodology: M.J.M., F.M., and M.J.C.; formal analysis and writing—original draft preparation: F.M.; writing—review and editing: M.J.M.; visualization and supervision: F.M. All authors have read and agreed to the published version of the manuscript.

**Funding:** This research received no external funding.

**Institutional Review Board Statement:** Not applicable.

**Informed Consent Statement:** Not applicable.

**Acknowledgments:** The authors want to thank the University of Malaga-Andalucia Tech, International Campus of Excellence, for its financial support of this paper.

**Conflicts of Interest:** The authors declare no conflict of interest.

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
