# Peer review of "The Upper Bound Theorem in Forging Processes: Model of Triangular Rigid Zones on Parts with Horizontal Symmetry†"

_applsci, doi:10.3390/app11010336_

Round 1
Reviewer 1 Report
The main problem of the manuscript is that it fails to present the novelty of the research work. A major revision is required to clarify what has been done in the literature on modelings and what is the contribution of this paper to the plastic deformation modeling.
Abstract looks more like an introduction. it is recommended to be shorter and be focused on the results of the paper, the novelty of the paper, and the methodology of the paper.
Introduction fails to present a comprehensive literature review and declare what the gap of knowledge is. For example, in line 51, it has been mentioned that "This method, a particular case of the SLD, is easier to implement and, nevertheless, provides quite acceptable solutions.". This is a very general statement without any explanation on why this method is easier to be implemented and what is the difference between this method and others which can make it a better one. or in line 58:"In the present article, the Upper Bound Method is approached through a Triangular Rigid Zones Model." Is it the first time somebody is employing this method? why are you using this method? elaboration on such these points are required in the introduction to better present the novelty and originality of the research work.
The results section is also very short and presents the results in a vague method which would make it again difficult to see why this method is better that other methods in the literature and what is new in this paper. also, the parameter p/2k should be defined.
Reviewer 2 Report
Please consult the attached document. Thanks.

Reviewer 3 Report
The paper present some interesting results extending the possibilities of application of the Upper Bound Theorem under the TRZ Method increasing the potentiality of the Modular approach.
The manuspcript is well organized and the conclusions well supported by the results.
However, some aspects need to be amendend before publication.
The abstract is quite long for a paper of 9 pages. Moreover, usually it is not preferred to insert citation in the abstract.
Linese 42-43: A brief description of these two methods should be provided. Some works which exploited these methods need to be mentioned.
Lines 45-48: you mention this other method , but without any reference to paper that involves it.
Lines 58-66: Triangular Rigid Zones Model is introduced here for the firts time but described just in line 77. Please, reorganize this part and stress more the novelty of your work respect to the letterature. In general the paper is poor of references.
Fig 1: the two figure should be disticly labeled (with letters a and b) and adequatelly described both in the text and in the label. All the angles and vectors in figure must be descrived.
Fig 2: It is not clear what the arrows and the "!" mean.
The conclusions are too generic and hasty. It would be preferable to rewrite them being more exaustive.
Line 290: In a scientific paper the "..." are not ammisible.
Round 2
Reviewer 1 Report
The authors have addressed the main concerns and the revised manuscript is acceptable for publication.
Reviewer 3 Report
The authors improved the paper, following my suggestions. The paper can now be accepted for publication in Applied Science.